# Training Characteristics and Power Profile of Professional U23 Cyclists throughout a Competitive Season

**DOI:** 10.3390/sports8120167

**Published:** 2020-12-17

**Authors:** Peter Leo, James Spragg, Dieter Simon, Justin S. Lawley, Iñigo Mujika

**Affiliations:** 1Department Sport Science, University Innsbruck, 6020 Innsbruck, Austria; justin.lawley@uibk.ac.at; 2Spragg Cycle Coaching, Exeter 03833, UK; james@spraggcyclecoaching.com; 3Training and Exercise Sciences, University of Applied Sciences Wiener Neustadt, 2700 Wiener Neustadt, Austria; dieter.simon@fhwn.ac.at; 4Department of Physiology, Faculty of Medicine and Nursing, University of the Basque Country, 48940 Leioa, Spain; inigo.mujika@inigomujika.com; 5Exercise Science Laboratory, School of Kinesiology, Faculty of Medicine, Universidad Finis Terrae, Santiago 8320000, Chile

**Keywords:** cycling, racing, volume, intensity, periodization, performance

## Abstract

Background: The purpose of this study was to investigate differences in the power profile derived from training and racing, the training characteristics across a competitive season and the relationships between training and power profile in U23 professional cyclists. Methods: Thirty male U23 professional cyclists (age, 20.0 ± 1.0 years; weight, 68.9 ± 6.9 kg; $\dot{V}$O_2max_, 73.7 ± 2.5 mL·kg^−1^·min^−1^) participated in this study. The cycling season was split into pre-, early-, mid- and late-season periods. Power data 2, 5, 12 min mean maximum power (MMP), critical power (CP) and training characteristics (Hours, Total Work, eTRIMP, Work·h^−1^, eTRIMP·h^−1^, Time_<VT1_, Time_VT1-2_ and Time_>VT2_) were recorded for each period. Power profiles derived exclusively from either training or racing data and training characteristics were compared between periods. The relationships between the changes in training characteristics and changes in the power profile were also investigated. Results: The absolute and relative power profiles were higher during racing than training at all periods (*p* ≤ 0.001–0.020). Training characteristics were significantly different between periods, with the lowest values in pre-season followed by late-season (*p* ≤ 0.001–0.040). Changes in the power profile between early- and mid-season significantly correlated with the changes in training characteristics (*p* < 0.05, r = −0.59 to 0.45). Conclusion: These findings reveal that a higher power profile was recorded during racing than training. In addition, training characteristics were lowest in pre-season followed by late-season. Changes in training characteristics correlated with changes in the power profile in early- and mid-season, but not in late-season. Practitioners should consider the influence of racing on the derived power profile and adequately balance training programs throughout a competitive season.

## 1. Introduction

The power profile [1,2,3] and training characteristics [4,5,6,7,8,9] of professional cyclists have been well documented in previous research. Depending on the rider type (sprinter, time trialist, allrounder or climber), professional cyclists require high absolute and relative power outputs over short (1 s to 5 min) to long durations (5 min to >4 h), at various levels of acute and chronic fatigue [1,2,3,10,11,12]. Differences in race topography (e.g. flat, semi- and high-mountainous stage profiles) [1,13] and race category [14] have been shown to influence the power profile derived from racing in professional cyclists. Power profiling can be used to predict race performance, especially during mountainous races, and can distinguish between different rider levels (e.g., U23 vs. professional) and rider types (e.g., allrounders, domestiques, general classification contenders) [12]. However, while it has been shown that the power profile can be derived from laboratory tests [2] and field tests, [3] it is unclear if the power profile can be derived from training data alone. The primary goal of training is to induce improvements in the power profile by prescribing work bouts based on the athlete’s strength and weaknesses. Therefore, training should be manipulated accordingly to bring about such improvements. Training and racing can be quantified through completed hours or distance, and using external (power output, total work) and internal (heart rate, rating of perceived exertion) training load measures [5,6,15]. Recent research demonstrated that training load can also be quantified using ratios of volume and intensity [16]. Additionally, the organization of training can be described using training intensity distribution (TID) [17]. Training volume and TID have been documented in U23 [18] and professional cyclists [5,8,9,19], but the relationship between changes in training characteristics and changes in the power profile has not yet been investigated. 

Therefore, the first aim of this study was to compare the power profile between training and racing to assess the differences of the power profile derived from training data. The second aim was to assess the variation in the training characteristics across a competitive season and to investigate the relationship between changes in training and changes in the power profile of professional U23 cyclists. 

## 2. Materials & Methods

### 2.1. Participants 

Thirty male U23 professional cyclists participated in this study (age, 20.0 ± 1.0 years; height, 182.6 ± 6.1 cm; weight, 68.9 ± 6.9 kg; $\dot{V}$O_2max_, 73.7 ± 2.5 mL·kg^−1^·min^−1^). All participants were members of a UCI Continental U23 development team during the cycling season(s) analyzed. Rider type classification was as follows: allrounders (*n* = 21) and climbers (*n* = 9) [20]. Recruitment was based on voluntary interest. In cases of any prolonged illness, injury (defined as no race days in a given period), or termination of cycling career, participants were excluded from all analysis. If a cyclist was included in the analysis over three consecutive years, he was treated as a separate participant for each year due to varying body mass and performance capacity. 

Informed written consent was obtained after each participant was given a verbal and written explanation of the experimental protocol and fully understood the possible risks involved in taking part in the study. The study protocol was approved (ID 382019) by the Ethical Review Board at the University of Innsbruck and followed the principles as set out in the declaration of Helsinki. 

### 2.2. Design

Power data files were collected from the participants during every training and racing session of a competitive season for 3 consecutive years. Each season was split into 4 periods: pre-season (November to January), early-season (February to April), mid-season (May to July) and late-season (August to October). 

Anthropometric data were collected in pre-season in conjunction with laboratory measures and at a randomized point within all other periods. During pre-season, participants performed both a laboratory-based graded incremental exercise test (GXT) and a critical power (CP) test consisting of 2, 5 and 12 min maximal efforts as per Leo et al. [11]. 

Throughout the season, 2, 5, and 12 min mean maximum power (MMP) data from each period were identified from power output files to produce a power profile for each cyclist. CP and work above CP (W′) estimates were derived from MMP values for training and racing.

Data were then processed in three ways: (1) differences in the absolute and relative power profile metrics; 2, 5 and 12 min MMP, CP and W′ values (absolute only) between training and racing for each period; (2) differences in training characteristics between periods; (3) relationship between the changes in the power profile metrics and the changes in training characteristics for each period. 

### 2.3. Methodology 

#### 2.3.1. Laboratory Testing

Participants were asked to avoid any exhaustive exercise 24 h prior to the test. They were also encouraged to appear in a fully rested, hydrated and fueled state. 

Open circuit spiroergometry with a breath by breath technique (ZAN600, nSpire Health GmbH, Oberthulba, Germany) was used. For the GXT, volume and flow were calibrated with a 1 L syringe before each trial. Gas analyzer calibration was completed before each measurement according to the manufacturer’s recommendations (4.9 Vol% CO_2_, 15.9 Vol% O_2_, 79.2 Vol% N_2_, nSpire Health GmbH, Oberthulba, Germany). All participants continuously wore a facemask and breathed through a flow sensor (FlowSensor Type II, nSpire Health GmbH, Oberthulba, Germany). $\dot{V}$O_2max_ was defined as the highest 30 s rolling average achieved before volitional exhaustion. The first ventilatory threshold (VT1) was defined as the point where the ventilation rate (VE) increased compared to $\dot{V}$O_2_ (VE/$\dot{V}$O_2_). The second ventilatory threshold (VT2) was defined as the onset of hyperventilation during the GXT [21], with an increase in VE compared to the volume of carbon dioxide ($\dot{V}$CO_2_) release, known as the VE/$\dot{V}$CO_2_ ratio. Continuous recordings of heart rate (HR) were performed via short range telemetry with a 1 Hz sampling rate (V800, Polar Electro Oy, Kempele, Finnland). The HR values corresponding to VT1 and VT2 were also determined. 

GXT was performed in a controlled environment (temperature, 19–22 °C) on the participant’s personal road bike (Alto Prestige, KTM Fahrrad GmbH, Mattighofen, Austria) mounted on an electromagnetically braked stationary trainer with a 1 Hz sampling rate (Cyclus2, RBM Elektronik-automation GmbH, Leipzig, Germany). The following GXT protocol was applied: initial workload 150 W, increment 20 W·min^−1^. In the case of a last uncompleted stage, maximal power output (P_max_) was calculated as per Kuipers et al. [22].
(1)$$P_{\max}=\;\mathrm{PL}\;+\left(\frac{t}{60}\times \;20\right)$$

Equation (1), P_max_ = maximum power output, PL = final completed stage in watts, *t* = associated time for the uncompleted work stage in seconds. 

#### 2.3.2. CP Test Protocol 

The CP test was carried out in the field during pre-season within two weeks of the initial laboratory visit on a standardized uphill climb with an average gradient of 5.5%, on two consecutive days, with an ambient temperature of 15–20 °C. 

The CP test consisted of 2, 5 and 12 min maximum efforts in randomized order. The 2 and 5 min efforts were performed on the same day interspersed by 30 min active recovery in which athletes were instructed to ride at a rating of perceived exertion (RPE) <2 out of 10. Prior to each effort the participants were encouraged to produce the highest possible workload and asked to maintain a cadence between 80 and 100 revolutions per minute (rev·min^−1^).

The inverse of time model, using a least sum of squares linear regression analysis, was used to derive CP and W′. The intercept of the regression line with the y axis represented CP and the slope W′. The following equation was applied:(2)$$P\;=\;W^{\prime }\;\times \frac{1}{t}\;+\;\mathrm{CP}$$

Equation (2), P = power output (W), *t* = duration of field test (s).

#### 2.3.3. Power Output 

Power output in the field was recorded using a standardized crank system (SRAM Red, Quarq, Spearfish, SD, USA) with a 1 Hz sampling rate, monitored on a portable head unit device (Garmin Edge 520, Schaffhausen, Switzerland). A static calibration of the power meter was undertaken prior to the laboratory visit in the pre-season according to Wooles et al. [23]. Participants were instructed to perform a “zero-offset” before each training or racing session. 

#### 2.3.4. Training Characteristics

Total accumulated cycling duration in both training and racing (Hours) was recorded for each period. External workload was quantified via Total Work (session duration (s) multiplied by the power output with a 1 Hz sampling rate). Internal workload was quantified using a five-zone model as per Edwards’ training impulse [24] (eTRIMP): zone 1, 50–59% HR_peak_; zone 2, 60–69% HR_peak_; zone 3, 70–79% HR_peak_; zone 4, 80–89% HR_peak_; and zone 5, 90–100% HR_peak_. HR_peak_ was defined as the highest HR recorded during early-, mid- or late-season period or during the GXT in pre-season. eTRIMP was then calculated by multiplying a zone-specific weighting factor (zone 1 = 1, zone 2 = 2, zone 3 = 3, zone 4 = 4, zone 5 = 5) by the total time accumulated in that zone. Workload ratios for external (Total Work) and internal (eTRIMP) workloads were divided by Hours for each period (Work⋅h^−1^ and eTRIMP⋅h^−1^ respectively). Total time accumulated at a HR below that corresponding to VT1 (Time_<VT1_), between VT1 and VT2 (Time_VT1-2_) and above VT2 (Time_>VT2_) were analyzed for each period. The number of race days was also recorded for each period. 

#### 2.3.5. Data Analysis

Absolute and relative training and racing 2, 5 and 12 min MMP values during every period were identified using a cycling software platform (WKO5 Build 562, TrainingPeaks LLC, Boulder, CO, USA). Each MMP output was manually checked for data spikes. The MMP values for each period were used to estimate CP and W′ for the relevant period for training and racing by applying the inverse of time CP model, using a least sum of squares linear regression analysis. 

Training characteristics including Hours, Total Work, eTRIMP, Work⋅h^−1^, eTRIMP⋅h^−1^, Time_<VT1_, Time_VT1-2_ and Time_>VT2_ were computed for each period. 

Delta values (∆) for the power profile and the training characteristics were derived. For comparisons between power profile, values between early- and pre-season CP test were used due to there being no racing in pre-season from which to derive values. 

#### 2.3.6. Statistical Analyses 

All values are expressed as mean ± standard deviation and or mean difference (∆). Normal distribution was tested using the Shapiro–Wilk test (*p* > 0.05). Statistical significance was established at *p* ≤ 0.05, two-tailed.

Differences in the power profile including absolute and relative MMP values, CP and W′ parameter estimates were compared between training and racing for each period using a one-way repeated measure analysis of variance (ANOVA). Differences in body mass and training characteristics between periods were also assessed using a repeated ANOVA. In the case of significance, a post-hoc Holm correction was applied for both. The repeated measures ANOVA was also applied if the data were not normally distributed, as was shown appropriate by Norman [25]. 

Pearson product correlation was used to investigate the relationship in changes between the power profile and training characteristics. 

All statistical analyses were completed using JASP statistics software (version 0.13.1 for Mac OS, JASP Team, Amsterdam, The Netherlands). All graphs and figures were created using GraphPad Prism (version 8.0.0 for Mac OS, GraphPad Software, San Diego, CA, USA).

## 3. Results

Descriptive data of the laboratory GXT and field CP tests are presented in Table 1. 

### 3.1. Power Profile

Absolute 2, 5 and 12 min MMP were higher in racing compared to training for early- (∆ = 33 W, *p* ≤ 0.001; ∆ = 23 W, *p* ≤ 0.001; ∆ = 23 W, *p* ≤ 0.001), mid- (∆ = 23 W, *p* = 0.005; ∆ = 26 W, *p* ≤ 0.001; ∆ = 28 W, *p* = 0.020) and late-season (∆ = 31 W, *p* ≤ 0.001; ∆ = 27 W, *p* ≤ 0.001; ∆ = 27 W, *p* ≤ 0.001). Absolute CP was also higher in racing compared to training for early- (∆ = 21 W, *p* ≤ 0.001), mid- (∆ = 29 W, *p* ≤ 0.001) and late-season (∆ = 27 W, *p* ≤ 0.001) (Figure 1). No significant differences were found for W′ between racing and training for either early-, mid- or late-season (*p* ≥ 0.05). 

Relative 2, 5 and 12 min MMP were higher in racing compared to training for early- (∆ = 0.4 W⋅kg^−1^, *p* ≤ 0.001; ∆ = 0.3 W⋅kg^−1^, *p* ≤ 0.001; ∆ = 0.3 W⋅kg^−1^, *p* ≤ 0.001), mid- (∆ = 0.3 W⋅kg^−1^, *p* = 0.005; ∆ = 0.4 W⋅kg^−1^, *p* ≤ 0.001; ∆ = 0.4 W⋅kg^−1^, *p* ≤ 0.001) and late-season (∆ = 0.4 W⋅kg^−1^, *p* ≤ 0.001; ∆ = 0.5 W⋅kg^−1^, *p* ≤ 0.001; ∆ = 0.4 W⋅kg^−1^, *p* ≤ 0.001). Relative CP was also higher in racing compared to training for early- (∆ = 0.3 W⋅kg^−1^, *p* ≤ 0.001), mid- (∆ = 0.4 W⋅kg^−1^, *p* ≤ 0.001) and late-season (∆ = 0.4 W⋅kg^−1^, *p* ≤ 0.001) (Figure 2). Body mass was the lowest in late-season compared to pre- (∆ = 0.8 kg, *p* = 0.031), early- (∆ = 1.1 kg, *p* ≤ 0.001) and mid-season (∆ = 1.0 kg, *p* = 0.003).

### 3.2. Training Characteristics

Training characteristics are presented in Table 2. 

Hours were lower in pre-season compared to early- (∆ = 34 h, *p* ≤ 0.001) and mid-season (∆ = 52 h, *p* ≤ 0.001) but higher than in late-season (∆ = 17 h, *p* = 0.027); Hours were higher in mid-season than in early-season (∆ = 17 h, *p* = 0.002); and Hours were lower in late-season than in early- (∆ = 51 h, *p* ≤ 0.001) and mid-season (∆ = 69 h, *p* ≤ 0.001).

Total work was lower in pre-season compared to early- (∆ = 42,319 kJ, *p* = 0.002) and mid-season (∆ = 57,477, *p* ≤ 0.001) and was lower in late-season than in early- (∆ = 35,286 kJ, *p* = 0.002) and mid-season (∆ = 50,444, *p* ≤ 0.001). Work⋅h^−1^ was lower in pre-season compared to early- (∆ = 129 kJ⋅h^−1^, *p* = 0.034), mid- (∆ = 145 kJ⋅h^−1^, *p* = 0.034) and late-season (∆ = 112 kJ⋅h^−1^, *p* = 0.015). 

eTRIMP was lower in pre- compared to early- (∆ = 5878 arbitrary unit (A.U.), *p* = 0.003) and mid- (∆ = 7558 A.U., *p* ≤ 0.001) but higher than in late-season (∆ = 6152 A.U., *p* = 0.003), and was lower in late- than in early- (∆ = 12,030 A.U., *p* ≤ 0.001) and mid-season (∆ = 13,710 A.U., *p* ≤ 0.001). eTRIMP⋅h^−1^ was lower in late- compared to pre- (∆ = 23.6 A.U.⋅h^−1^, *p* = 0.007) and early-season (∆ = 17.5 A.U.⋅h^−1^, *p* = 0.007). 

Time_<VT1_ was lower in pre- compared to early- (∆ = 10.1 h, *p* = 0.002) and late-season (∆ = 12.6 h, *p* ≤ 0.001) and lower in late- compared to early- (∆ = 10.1 h, *p* = 0.004) and mid-season (∆ = 12.6 h, *p* ≤ 0.001). 

Time_VT1-2_ was lower in pre- compared to early- (∆ = 20 h, *p* = 0.005) and mid-season (∆ = 21 h, *p* = 0.011) but higher than in late-season (∆ = 28.3 h, *p* = 0.002); it was lower in late- compared to early- (∆ = 48.5 h, *p* ≤ 0.001) and mid-season (∆ = 49.2 h, *p* ≤ 0.001). 

Time_>VT2_ was lower in pre- compared to early- (∆ = 3.5 h, *p* = 0.040) and mid-season (∆ = 5.1 h, *p* = 0.017) but higher than in late-season (∆ = 3.2 h, *p* = 0.040); it was lower in late- compared to early- (∆ = 6.8 h, *p* ≤ 0.001) and mid-season (∆ = 8.4 h, *p* ≤ 0.001). 

The number of race days was higher in mid- compared to early- (∆ = 6, *p* ≤ 0.001) and late-season (∆ = 6, *p* ≤ 0.001).

### 3.3. Relationship between Changes in Training Characteristics and Changes in Power Profile

The ∆ in 2 and 5 min MMP between early- and pre-season significantly correlated with ∆Work (r = −0.53, *p* = 0.002; r = −0.59, *p* ≤ 0.001, respectively, Figure 3A,B) and ∆Work⋅h^−1^ (r = −0.42, *p* = 0.019, Figure 3C). The ∆ in 12 min MMP and CP between early- and pre-season significantly correlated with race days (r = −0.44, *p* = 0.014; r = −0.40, *p* = 0.027, Figure 3D,E).

The ∆ in 2 and 12 min MMP between mid- and early-season significantly correlated with the ∆Time_<VT1_ (r = 0.41, *p* = 0.022; r = 0.42, *p* = 0.020—Figure 4A,B). The ∆ in CP between mid- and early-season significantly correlated with ∆Time_>VT2_ (r = 0.45, *p* = 0.012, Figure 4C).

## 4. Discussion

This study aimed to investigate differences in the power profile between training and racing, changes in training characteristics between different seasonal periods and correlations between these variables. The absolute and relative power profile was higher in racing compared to training for early-, mid- and late-season. Training characteristics were significantly different between periods and changes in these training characteristics significantly influenced the changes seen in the power profile. 

In previous work, Leo et al. [11] found no differences in the absolute power profile of U23 cyclists across a competitive season. Changes in the relative power profile were primarily due to varying body mass. Those changes in relative MMP and CP values are accompanied by a reduction in body mass the longer the season lasts due to accumulated race days and training volume. However, the authors did not differentiate between training and racing efforts. In the present study, a higher absolute (4.6–8.5%) and relative (4.2–8.4%) power profile was recorded during racing than training for all periods of the season. These findings suggest that power outputs recorded in training alone are not reflective of a true maximal power profile in U23 professional cyclists. Interval training sessions in cycling are commonly prescribed by power output, heart rate or perception of effort, and are rarely prescribed as maximal effort [26,27]. Recent research has shown that the accumulated time at or above 90% HR_max_ during interval training is sufficient to elicit cardiovascular adaptations such as increased cardiac output and stroke volume. Therefore efforts do not need to be maximal in nature to induce adaptations [27]. Leo et al. [11] also reported that efforts in racing might determine the power profile, as they found good agreement between CP derived from formal testing and field-derived MMP values. This confirmed previous findings by Karsten et al. [28]. In contrast, Pinot and Grappe concluded that MMP values derived from racing do not reflect a cyclist’s true maximum power profile [3,29]. A possible explanation for this is that efforts in racing may be influenced by race scenarios and team tactics [11], glycogen depletion [30] or model fitting [31]. Race topography and category, as well as short term fatigue in single day racing or accumulated fatigue in multi-stage racing, have also been shown to influence the power profile [12,14,32,33]. It has thus been recommended to verify field derived MMP values with a minimum of two maximum effort field tests per season (i.e. CP tests) for baseline comparisons. In summary, MMP and CP values derived from racing efforts in combination with a prior CP field test may offer the best way to longitudinally monitor the power profile in elite cyclists. 

Training characteristics were significantly different between periods. Indeed, both higher volume and intensity were seen in the early- and mid-season compared to pre- and late-season. From pre- to early-season, volume characteristics including Hours, Total Work and eTRIMP increased by 18.6 to 46.7%, while intensity metrics Work⋅h^−1^ (+24.3%) and eTRIMP⋅h^−1^ (−3.1%) diverged. In mid-season, volume characteristics including Hours, Total Work and eTRIMP were higher by 4.5 to 11.4% compared to early-season. Intensity metrics including Work⋅h^−1^ (+7.9%) and eTRIMP⋅h^−1^ (−4.4%) were again divergent. From mid- to late-season, both volume (−4.7 to −35.1%) and intensity (−5.6 to −58.5%) metrics were clearly declining. The number of race days was 53.8% higher during mid- compared to early-season, while there were 30% fewer race days in late- compared to mid-season. A possible explanation for the conflicting findings between volume and intensity metrics is that the participants’ heart rate responses were lowered either due to the accumulated training load and subsequent accumulated fatigue or improved cardiovascular adaptations [34,35]. The decline in training characteristics could be triggered by the residual fatigue accumulated during the entire season through training and racing [16,35], precluding athletes from maintaining or increasing training volume and or intensity. In contrast, the power profile was maintained throughout a competitive season, with some improvements in the relative power profile due to reduced body mass [11]. This may indicate that excessive fatigue negatively influences training characteristics rather than the power profile, and therefore performance, in the short term [36]. 

Research has shown that a polarized training intensity distribution may be the preferred training strategy in elite endurance athletes [37]. Stöggl and Sperlich reported a training intensity distribution of 68 ± 12% Time_<VT1_, 6 ± 8% Time_VT1-2_ and 26 ± 7% Time_>VT2_ in endurance athletes following a polarized training approach [17]. However, in the present study, training intensity distribution did not follow a polarized approach. The average distribution across the entire season was 17.4–19.4% in Time_<VT1_, 50.8–62.5% in Time_VT1–2_ and 8.0–9.3% in Time _> VT2_ and could thus be classified as a threshold training intensity distribution. We hypothesized that the high percentage of Time_VT1–2_ may have been due to the high number of race days, as in races athletes cannot control the power requirement; however, post hoc testing showed that there was no correlation between race days and Time_VT1-2_ for any period, nor did athletes with lower CP record more Time_VT1-2_. Therefore, the high percentage of total training Time_VT1-2_ may be due to this distribution being the coaches’ desired training distribution, or it was due to poor intensity zone discipline with athletes executing low intensity sessions too hard [38,39]. Another possibility is that the training zones used by coaches were not anchored, or were inaccurately anchored, to physiological thresholds. 

The relationship between changes in the power profile and training characteristics from pre- to early-season revealed that if the riders increased training load or race days too much, a decrease in the power profile occurred, as Total Work, Work⋅h^−1^ and race days negatively correlated with the power profile. This was evidenced by the previously discussed divergence between external and internal intensity metrics where Work⋅h^−1^ increased by +24.3% whereas eTRIMP⋅h^−1^ fell by 3.1%. From pre- to early-season there was a larger increase in training Time_VT1-VT2_ compared with Time_<VT1_. Previous work has shown that VT1 may represent a threshold intensity in relation to the level of fatigue induced in the autonomic nervous system [40]. 

Therefore, a practical recommendation may be that as racing is introduced in early-season, total work should not be further increased; to achieve this, a reduction in the intensity of the overall volume may be beneficial. This approach would also induce a shift towards a polarized training intensity distribution. This is evidenced in the relationship between the change in the power profile and training characteristics from early- to mid-season, where training Time_<VT1_ and Time_>VT2_ positively correlated with the changes in the power profile. Essentially, a shift towards a polarized training intensity distribution positively influenced the power profile, which has been shown to have a positive impact on race performance [12]. Interestingly, no relationship between the change in the power profile and training characteristics was found for mid- to late-season. As training load clearly decreased during late-season, it may not provide any predictive ability to monitor the power profile. It may be that in late-season, riders enter a maintenance phase whereby the minimum effective training dose [36] is used to maintain the power profile [11]. 

The authors are aware that the current study is not without limitations. Power output data could be influenced by stochastic and non-stochastic pacing patterns due to training or racing in mass start events as well as team strategies or race tactics [41]. Using fixed duration for MMP values could over- or underestimate the maximum capable power output, as it could be part of a longer effort. MMP values could be recorded right at the beginning, middle or end of a period, but as shown in previous research [11,42], the power profile during a competitive season remains relatively consistent. Regarding training documentation, only cycling-specific workouts were quantified, as they represented the majority of training, However the cyclists could have still completed additional cross training activities (XC-skiing, ski touring, running, strength and conditioning) during pre-season which could not be assessed in this study due to different training devices and training log documentation.

In conclusion, the current study found a higher absolute and relative power profile during racing compared to training across a competitive season. Training characteristics in volume and intensity increased from pre- to early- until mid-season, while in late-season a reduction in training volume could be seen. Changes in training characteristics are predictive for changes in the power profile for pre- until mid-season. Interestingly, although training characteristics, i.e. volume, decline in late-season, the riders could maintain their power profile without seeing a reduction compared to previous periods.

## Figures and Tables

**Figure 1 sports-08-00167-f001:**
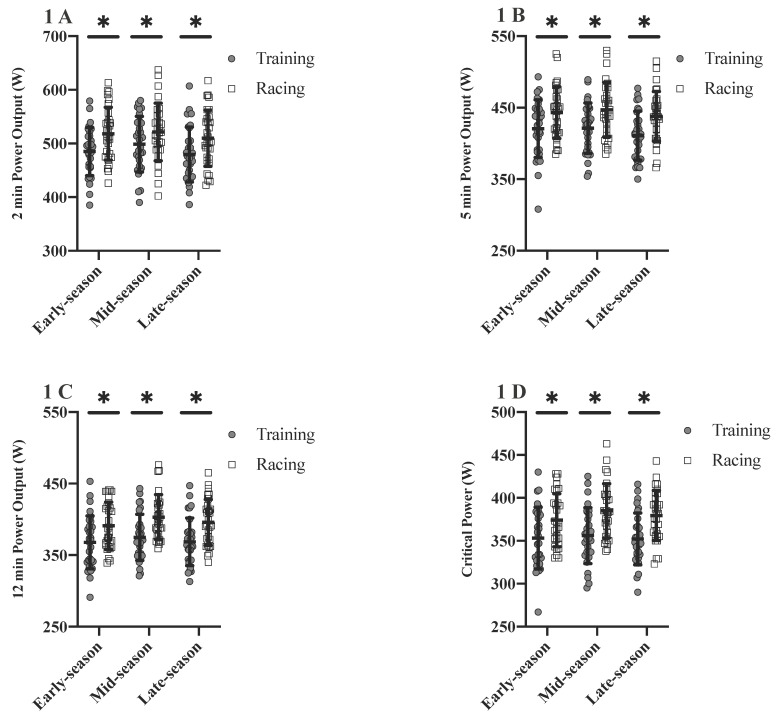
Differences in the absolute power profile between training and racing across periods; (**A**)—2-min power output, (**B**)—5-min power output, (**C**)—12-min power output, (**D**)—critical power. * significantly different between training and racing (*p* ≤ 0.05).

**Figure 2 sports-08-00167-f002:**
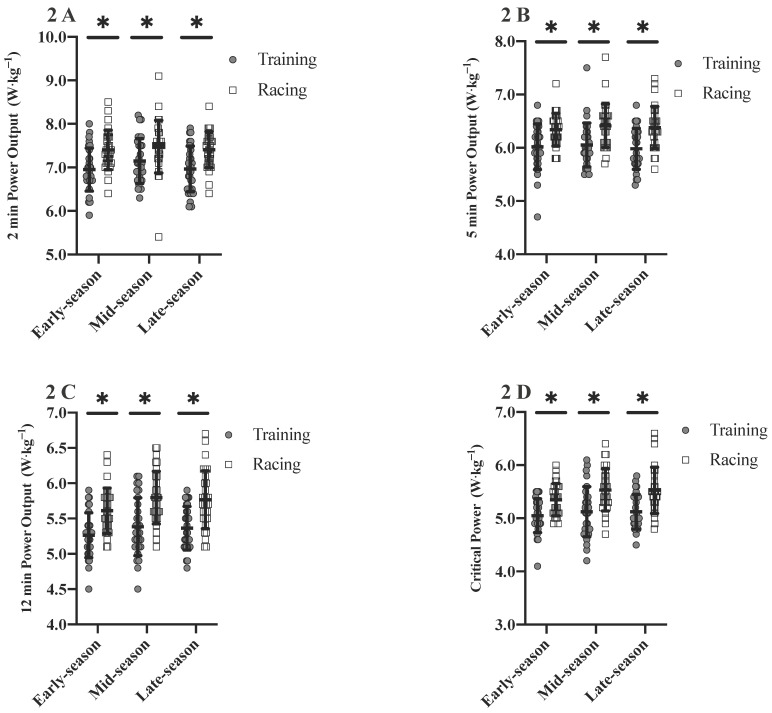
Differences in the relative power profile between training and racing across periods; (**A**)—2 min relative power output, (**B**)—5 min relative power output, (**C**)—12 min relative power output, (**D**)—relative critical power. * significantly different between training and racing (*p* ≤ 0.05).

**Figure 3 sports-08-00167-f003:**
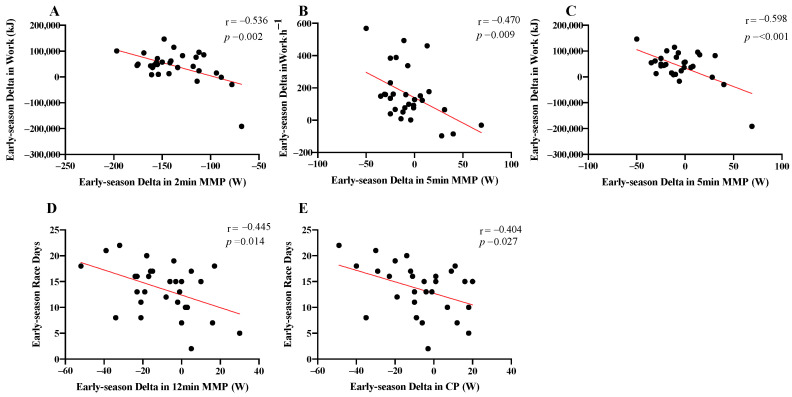
Relationship between the change in the power profile and training characteristics for the ∆ early- vs. pre-season. MMP—mean maximum power, CP—critical power; (**A**)—Work and 2 min MMP, (**B**)—Work⋅h^−1^ and 5 min MMP, (**C**)—Work and 5 min MMP, (**D**)—Race Days and 12 min MMP, (**E**)—Race Days and CP.

**Figure 4 sports-08-00167-f004:**
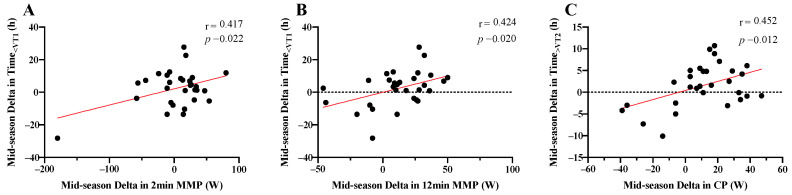
Relationship between the change in power profile and training characteristics for ∆ mid- vs. early-season. MMP—mean maximum power, CP—critical power, Time_<VT1_—time below the first ventilatory threshold, Time_>VT2_—time above the second ventilatory threshold; (**A**)—Time_<VT1_ and 2 min MMP, (**B**)—Time_<VT1_ and 12 min MMP, (**C**)—Time_>VT2_ and CP.

**Table 1 sports-08-00167-t001:** Physiological characteristics of professional U23 cyclists from the GXT (graded incremental exercise test) and CP (critical power) test (mean ± SD).

Variables	Absolute Values	Relative Values
P_max_	458 ± 38 [W]	6.6 ± 0.4 [W⋅kg^−1^]
$\dot{V}$O_2max_	5076 ± 424 [mL⋅min^−1^]	73.7 ± 2.5 [mL⋅kg^−1^⋅min^−1^]
VT1	256 ± 22 [W]	3.8 ± 0.3 [W⋅kg^−1^]
VT2	367 ± 38 [W]	5.3 ± 0.5 [W⋅kg^−1^]
CP	382 ± 33 [W]	5.5 ± 0.4 [W⋅kg^−1^]
W′	17.8 ± 3.6 [kJ]	n/a

GXT—graded incremental exercise test, P_max_—maximum power output; $\dot{V}$O_2max_—maximum oxygen uptake; VT1—first ventilatory threshold; VT2—second ventilatory threshold; CP—critical power; W′—work above critical power

**Table 2 sports-08-00167-t002:** Training characteristics of professional U23 cyclists across periods (mean ± SD).

Variables	Pre-Season	Early-Season	Mid-Season	Late-Season
Hours (h)	167 ± 46	202 ± 28 *	219 ± 26 *^,#^	150 ± 36 *^,#,†^
Total Work (kJ)	90.507 ± 45.622	132.825 ± 36.738 *	147.983 ± 33.497 *	97.539 ± 35.832 ^#,†^
Work⋅h^−1^(kJ⋅h^−1^)	529 ± 182	658 ± 143 *	674 ± 119 *	642 ± 142 *
eTRIMP (A.U.)	31.477 ± 9.543	37.356 ± 6.416 *	39.036 ± 8.007 *^,#^	25.325 ± 7.960 *^,#^
eTRIMP⋅h^−1^(A.U.⋅h^−1^)	192 ± 33	186 ± 28 *	178 ± 31	168 ± 33 *^,#^
Time_<VT1_ (h)	29.2 ± 11.7	39.4 ± 19.1 *	41.8 ± 16.4	29.2 ± 15.8 *^,#,†^
Time_VT1-2_ (h)	104.5 ± 36.0	124.7 ± 21.0 *	125.4 ± 27.0 *	76.2 ± 29.4 *^,#,†^
Time_>VT2_ (h)	15.3 ± 8.0	18.9 ± 5.2 *	20.4 ± 6.6 *	12.0 ± 5.6 *
Race Days	n/a	13 ± 5 ^#^	20 ± 6	14 ± 7 ^#^

Hours—training hours, eTRIMP—Edwald’s training impulse; VT—ventilatory threshold, * significantly different from pre-season; ^#^ significantly different from early-season, ^†^ significantly different from mid-season (*p* ≤ 0.05).

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
