# Peer review of "Training Characteristics and Power Profile of Professional U23 Cyclists throughout a Competitive Season"

_sports, 2020, doi:10.3390/sports8120167_

Round 1
Reviewer 1 Report
This study was to compare the power profile between training and racing season, and analyzed the relationship between changes in training and changes in the power profiles of professional U23 cyclists. Therefore these results are a good information for the optimal training program periodization.
Overall it is well written, however, It is expected that it will be more helpful if discussion on the relationship between changes in performance and power profile during the actual season is added.
Author Response
This study was to compare the power profile between training and racing season, and analyzed the relationship between changes in training and changes in the power profiles of professional U23 cyclists. Therefore these results are a good information for the optimal training program periodization.
We very much appreciate the positive comments and the critical yet constructive feedback on our manuscript.
Overall it is well written, however, It is expected that it will be more helpful if discussion on the relationship between changes in performance and power profile during the actual season is added.
Thank you for this comment. As shown in our previous research the power profile stays quite robust throughout the season, with no significant changes in absolute MMP and CP values. The changes in relative MMP and CP values are primarily due to a reduction in body mass as the season progresses, due to accumulated race days and training volume - see L256-257.
For this reason, we added that body mass was lowest in late-season period - see L198-200.
We also saw that as soon as racing is introduced in early-season, total work should not be further increased; to achieve this, a reduction in the intensity of the overall volume may be beneficial. This approach would also induce a shift towards a polarized training intensity distribution. This is evidenced in the relationship between the change in the power profile and training characteristics from early- to mid-season, where training Time<VT1 and Time>VT2 positively correlated with the changes in the power profile - see L316-321.

Reviewer 2 Report
see attachment

Author Response
This study assesses the power profile obtained from training and racing, training characteristics throughout the different phases of the season and correlation between changes in training and power profiles in professional cyclists under 23. The manuscript addresses a question of significant importance and is generally well written. There are some minor concerns that need to be addressed before publication.
We very much appreciate the positive comments and the critical yet constructive feedback on our manuscript. We have responded to each comment from the reviewer and believe the resulting changes have strengthened the manuscript.
1.Please elaborate on what the distribution of cycling disciplines was for the riders included in this study (e.g. uphill, flat, sprint, time-trial). Also, explain in more detail whether and how findings for training and racing power profiles are influenced by the different cycling disciplines.
Thank you for this comment. In the U23 level a specialization into a distinct rider type is not recommended, as they are still in their maturation to professional cyclists. However, we have added that rider type classification into allrounders (N=21) and climbers (N=9) was based on Giorgi et al. [reference #41 in the manuscript] - see L69-70.
2.The authors mention that the validity of the power profile from training data is assessed (L59). Please explain in more detail how this validity has been tested, as I could not find any details on this in the statistics and results sections.
Thank you for this comment. The term validity is misleading here, as we investigated the differences in the power profile. We therefore added that the aim of this study was to compare the power profile between training and racing to assess the differences of the power profile derived from training data - see L61.
3.How did the riders’ weight change throughout the season? Was this related to differences in the power profile obtained from racing and training?
Thank you for this comment. We have added the changes in body mass between periods to our result section. Body mass was the lowest in late- compared to pre- (∆=0.8 kg, p=0.031), early- (∆=1.1 kg, p≤0.001) and mid-season (∆=1.0 kg, p=0.003) - see L198-200.
We agree that changes in body mass contribute to changes in the relative power profile throughout a competitive season, as we have already reported in one of our previous studies, which is currently accepted for publication. Concerning the current study, the same body mass for each period was used for training and racing, thus the influence of a reduction in body mass to improvements in the relative power profile between racing and training can be neglected.
4.Lowest training values were observed in the pre-season, which is remarkable as the riders have more time available for training (since they do not race). Could the authors explain why this was not unexpected (L269-270)?
Thank you for this comment. The team, as well as most of the riders are based in Austria, where the winter months are suboptimal for cycling. Only cycling specific workouts were quantified, as they represented the majority of training, but the cyclists could have also completed additional cross training activities (XC-skiing, ski touring, running, strength and conditioning) during pre-season. This, unfortunately, could not be assessed in this study due to different training devices and training log documentation - see 336-340.
5.The results section sums up a lot of detailed findings. It may help readers if concluding sentences were added or paragraphs were connected using linking words.
Thank you for this comment. All major results of this study are summarized in a separate conclusion paragraph - see 341-347.
6.A concluding paragraph or conclusion section seems to be missing(L326).
Thank you for this comment. As mentioned above a conclusion paragraph has been added - see 341-347.
Specific comments
L21. Please explain abbreviations upon first mention in the text.
Thanks for this comment. Mean maximum power (MMP) and critical power (CP) has been added - see L21.
L50-51. Consider rephrasing.
Thank you for this consideration. We changed it as follows: However, while it has been shown that the power profile can be derived from laboratory tests and field tests, it is unclear if the power profile can also be derived from training data alone. The primary goal of training is to induce improvements in the power profile by prescribing work bouts based on the athlete’s strengths and weaknesses. Therefore, training should be manipulated accordingly to bring about such improvements - see L48-53.
L51. Check spelling of “quantified”
Thank you; we corrected the spelling - see L53.
L83. Consider changing into “12-min maximal efforts”
Thank you; we changed it accordingly - see L87.
L85. Consider changing into “power output files”
Thank you; we changed it accordingly - see L89.
L85. Was power output measured with the same equipment in all riders?
Thank you for this comment. Power output in the field was recorded using a standardized crank - system (SRAM Red, Quarq, Spearfish, South Dakota, USA) with a 1 Hz sampling rate and monitored on a portable head unit device (Garmin Edge 520, Schaffhausen, Switzerland) - see L135-137.
L87. Consider changing “interrogated”.
Thank you; we changed it to “processed” - see L91.
L101. Consider changing into “30-s rolling average”.
Thank you; we changed it accordingly - see L105.
L102-103. Please explain whether VT1 was only determined by VE/VO2 and not by the V-slope method and elaborate in more detail what is meant with onset of hyperventilation. I would assume this is the increase in VE/VCO2.
Thank you for this comment. We have added that the first ventilatory threshold (VT1) was defined as the point where the ventilation rate (VE) increased compared to O2 (VE/O2). The second ventilatory threshold (VT2) was defined as the onset of hyperventilation during the GXT[1], with an increase in VE compared to the volume of carbon dioxide (CO2) release known as the VE/CO2 ratio - see L106-109.
L121. Do the authors mean that RPE dropped to 2 out of 10 within the 30-min active recovery?
Thank you for this comment. The riders where instructed to continue with active recovery between inter-trial recovery, guided by an RPE of 2 out of 10 - see L125.
L136. How is total work defined?
Thank you for this question. We added that external workload was quantified via Total Work (duration multiplied by the power output with a 1 Hz sampling) - see L141-142.
L139. What is meant with “follow-up period”? Is this throughout the season?
Thank you for this question. We changed it to HRpeak was defined as the highest HR recorded during that period or the GXT - see L145.
L149-151. Please explain in more detail how this was done.
Thank you for this comment. Activity files were downloaded from the riders’ head unit to a database. From this database the 2, 5 and 12 min MMP values were derived for each athlete for each period. The inverse of time model, using a least sum of squares linear regression analysis, was used to derive CP and W´ from the MMP values. The intercept of the regression line with the y axis represented CP and the slope W´. The following equation was applied: P =W' ×1/t +CP - see L129-131 and L152-156.
L259. Consider changing into “field-derived MMP values”.
Thank you; we changed it accordingly - see L266.
L261-262. Is there a minimal number of racing days that is required to overcome the confounding effects of these influences?
Thank you for this question. Every athlete involved in this study was racing in each period, otherwise he would be excluded from the data analysis - see L70-72. The lowest number of races by one athlete for late-season period was 7 races, which was nevertheless sufficient to record higher MMP values during racing than training.
L291-293. Something seems to be wrong here. With the current percentages no rider could have reached 100%. Please explain.
Thank you for this question. The percentage values mentioned here are calculated from the Time in Zone <VT1, VT1-2 and >VT2 across early-, mid- and late-season periods. These percentage values are not linked to an individual rider per se, as the idea was to represent the overall trend of training intensity distribution.
Fig1 & 2. Significance symbols are missing.
Thank you for this comment. Significance symbols (*) were added to figures 1 and 2.
Figure 1. Differences in the absolute power profile between training and racing across periods, *significantly different between training and racing (p≤0.05).
Figure 2. Differences in the relative power profile between training and racing across periods, *significantly different between training and racing (p≤0.05).
Fig3 & 4. Were these values of the power profile obtained from racing or training (since there is no racing in the pre-season)?
Thank you for this question. We only used MMP values of racing for early-, mid- and late-season, as well as the formal CP field test for pre-season. This approach was already used in one of our previous papers, which is currently in press. We showed that valid predictions can be derived from formal CP field testing – therefore these values were used for pre-season, when there were no races.
Also please alter the labels so that it is known that the delta refers to early season vs. pre-season (Fig3) and mid-season vs. early-season (Fig4)
Thank you for this comment. We added Figure 3. Relationship between the change in the power profile and training characteristics for the ∆ early- vs. pre-season. We added Figure 4. Relationship between the change in power profile and training characteristics for ∆ mid- vs. early-season - see L238-239 and L244-246.
Table2. Please add the thousands separator
Thank you; we changed it accordingly - see L206.

Reviewer 3 Report
Estimates authors,
Although the study is correctly designed and carried out, the significance of the results and conclusions were low, due to the obviousness of them.
Most of the trainers and coaches know it so the usefulness of the results as a practical level is low.
MATERIALS AND METHODS
line 74:
The study protocol was approved by the Ethical Review Board at the University of Innsbruck
- Do you have the code of the ethical committee?
Why did you choose 2, 5, 12 min MMP and absolute CP to measure power during training and what is more, during racing?
DISCUSSION
Could you explain the causes of the differences in the results between 2, 5, 12 min MMP and CP? All of them are higher in racing than in training, (absolute and relative). For example: Any differences between 2 and 12 min-? Why?
LIMITATIONS:
Injuries?
Classification/ranking?
Team strategies? Leader/gregarious?
Same performance goals for all the sample?
Author Response
Estimates authors,
Although the study is correctly designed and carried out, the significance of the results and conclusions were low, due to the obviousness of them.
Most of the trainers and coaches know it so the usefulness of the results as a practical level is low.
We very much appreciate the positive comments and the critical yet constructive feedback on our manuscript. As three authors of our team are working in professional cycling, we are aware of the fact that some of this information may not surprise experienced coaches. However no published data could confirm this phenomenon until now. Therefore, we feel that this scientific contribution adds a valuable context to that field and strengthens the knowledge base about power profiling for practitioners.
MATERIALS AND METHODS
line 74:
The study protocol was approved by the Ethical Review Board at the University of Innsbruck
Do you have the code of the ethical committee?
Thank you for this comment. The code for the ethical committee was sent to the editorial office at the time when the manuscript was submitted.
Why did you choose 2, 5, 12 min MMP and absolute CP to measure power during training and what is more, during racing?
Thank you for this question. In previous research, which is currently accepted for publication elsewhere, we found that CP derived from 2, 5 and 12 mim MMP during racing and a formal CP field test can be used interchangeably to describe the power profile of U23 riders. From our data it seems that the riders find another gear during racing (motivation, race situation) compared to training, where most efforts are performed in a controlled environment with power targets prescribed for interval training.
DISCUSSION
Could you explain the causes of the differences in the results between 2, 5, 12 min MMP and CP? All of them are higher in racing than in training, (absolute and relative). For example: Any differences between 2 and 12 min-? Why?
Thank you for these questions. CP is derived from 2, 5 and 12 min MMP with a linear regression approach. The inverse of time model, using a least sum of squares linear regression analysis, was used to derive CP and W´ from the MMP values. The intercept of the regression line with the y axis represented CP and the slope W´. The following equation was applied: P =W' ×1/t +CP - see 129-131. Generally speaking, shorter maximum power outputs always indicate higher power values than the longer efforts due to the curvilinear relationship of power output and duration.
LIMITATIONS:
Injuries?
Thank you for this concern. If an athlete suffered from injury he would be excluded from further analysis - see L70-72.
Classification/ranking?
Thank you for this question. No race results were used to characterize our participants.
Team strategies? Leader/gregarious?
Thank you for this comment. We are aware that team strategies and race tactics could have influenced power output during racing - see line 330.
Same performance goals for all the sample?
Thank you for this question. No special performance goals were set for conducting this study from the authors’ side. However, we agree that the riders could have applied individual performance goals during racing, which was not further investigated in this study.

Reviewer 4 Report
Does the introduction provide sufficient background and include all relevant references?
I believe that although the introduction is short, it is correct and uses current references.
Is the research design appropriate?
In my opinion, the research design is adequate
Are the methods adequately described?
I think the methods are adequately described.
Are the results clearly presented?
What does the number 11 in the above index on line 262 mean?
Are the conclusions supported by the results?
The manuscript does not include a conclusion. A conclusion must be included in the manuscript.
Author Response
We very much appreciate the positive comments and the critical yet constructive feedback on our manuscript. We have responded to each comment from the reviewer and believe the resulting changes have strengthened the manuscript.
Does the introduction provide sufficient background and include all relevant references?
I believe that although the introduction is short, it is correct and uses current references.
Thank you very much for this positive comment.
Is the research design appropriate?
In my opinion, the research design is adequate
Thank you very much for this positive comment.
Are the methods adequately described?
I think the methods are adequately described.
Thank you very much for this positive comment.
Are the results clearly presented?
What does the number 11 in the above index on line 262 mean?
Thank you for this comment. The journal’s reference style was adopted accordingly - see L269.
Are the conclusions supported by the results?
The manuscript does not include a conclusion. A conclusion must be included in the manuscript.
Thank you for this comment. We concluded that the current study found a higher absolute and relative power profile during racing compared to training across a competitive season. Training characteristics in volume and intensity increased from pre- to early- until mid-season, while in late-season a reduction in training volume could be seen. Changes in training characteristics were predictive of changes in the power profile for pre- until mid-season. Interestingly although training volume declined in late-season, the riders could maintain the power profile compared to previous periods - see L341-347.
